# The Effect of Alternating Monocular Instillation of 0.125% Atropine in Korean Children with Progressive Myopia

**DOI:** 10.3390/jcm13175003

**Published:** 2024-08-23

**Authors:** Ji Sang Min, Byung Moo Min

**Affiliations:** 1Department of Ophthalmology, The Institute of Vision Research, Yonsei University College of Medicine, Seoul 03722, Republic of Korea; jsmansae@naver.com; 2Woori Eye Clinic, Affiliated to the Department of Ophthalmology, Yonsei University College of Medicine, Daejeon 35229, Republic of Korea

**Keywords:** 0.125% atropine, alternating monocular instillation, progressive pediatric myopia, adverse event

## Abstract

**Objectives**: To identify the effect of alternating monocular instillation (AMI) of 0.125% atropine in Korean children with progressive myopia. **Methods**: This retrospective single-center study included 120 children with progressive myopia. A total of 60 children (mean age 9.2 ± 2.0 years) wearing glasses who received AMI of 0.125% atropine for one year were allocated to the treatment group. The remaining 60 children (mean age 9.2 ± 1.9 years) with the same refraction, SE, and axial length (AL) who did not receive any treatments except for wearing glasses were allocated to the control group. Ocular findings and the progression rate were compared between the groups pre- and post-treatment, and adverse events were investigated in the treatment group. **Results**: The mean spherical equivalent (SE) at baseline was −3.87 ± 1.55 D in the control group and −3.90 ± 1.56 D in the treatment group. Pre-treatment SE, age, and AL were similar between the groups; however, post-treatment SE and AL changes were smaller in the treatment group (−0.36 ± 0.46 D/y, 0.21 ± 0.20 mm/year in the treatment group vs. −1.02 ± 0.57 D/y, 0.51 ± 0.20 mm/year in the control group) (Ps < 0.001). The pre-treatment progression rate diminished in the treatment group compared to the control group after one year (*p* < 0.001), and the changes in pupil size under mesopic and photopic conditions in the treatment group increased by 0.03 ± 0.05 mm and 0.76 ± 0.90 mm, respectively. Regarding adverse events, a tingling sensation was noted in two patients (3.3%) in the treatment group. **Conclusions**: Alternating monocular 0.125% atropine eye drop instillation may be effective and suitable for progressive myopia in Korean children.

## 1. Introduction

Myopia is very common in Asia and is increasing in frequency, with an incidence of up to 90% [1,2,3,4]. Myopia progression occurs primarily due to axial elongation. High myopia can cause glaucoma, cataract formation, myopic macular degeneration, and retinal detachment in adults; thus, suppression of myopia progression is crucial [5,6].

The currently reported methods of suppressing myopia progression include bifocal spectacle lenses [7,8,9], peripheral defocusing lenses, contact lenses [10,11], overnight orthokeratology [12,13], multifocal soft contact lenses, outdoor activities, and pharmacological agents. Moreover, according to the ATOM I study, 1% atropine eye drop treatment was the most effective treatment for myopia suppression [14,15,16,17,18]. Although 1% atropine eye drop treatment is very effective in reducing spherical equivalent (SE) or axial length (AL) increases, it has side effects, such as phototoxic effects on the retina and lens, near vision blurring, photophobia, allergic reactions, and myopic rebound after the treatment is discontinued [16,19,20].

The ATOM II study reported that low-dose atropine (0.5%, 0.1%, and 0.01%) was associated with fewer side effects while being relatively effective in suppressing myopia progression [20]. In a systematic review and meta-analysis, 0.05% atropine was identified as the optimal dose [21]. Moreover, low-dose atropine (0.01%, 0.02%, 0.025%, and 0.05%) has been reported to be effective and safe [22,23,24,25,26,27]. The safety and efficacy of atropine treatment for slowing myopia progression in children were also reported in a 5-year review [27,28]. In November 2020, the Ministry of Food and Drug Safety of the Republic of Korea approved the use of 0.125% low-concentration atropine (only one dosage). However, only two studies have reported the results of administering 0.125% atropine eye drops to effectively suppress myopia progression. In these studies, 4.05% and 74% of patients, separately, experienced pupil dilation, photophobia, and disturbed near vision during the initiation of binocular 0.125% atropine treatment [28,29]. We postulated that alternating monocular instillation (AMI) of 0.125% atropine eye drops could reduce the atropine concentration in patients’ eyes, induce fewer side effects, and improve compliance for eye drop instillation. Therefore, this retrospective study aimed to evaluate the efficacy of alternating monocular 0.125% atropine eye drop instillation treatment in children with progressive myopia.

## 2. Materials and Methods

A clinic-based, retrospective efficacy and safety study was conducted at the Woori Eye Clinic (Daejeon, Republic of Korea). All children younger than 14 years of age presenting with progressive myopia were eligible for inclusion in the study. The inclusion criteria were SE ≤ −6.0 D and a SE progression rate ≥ 1 D in the previous year, anisometropia under 2 diopters under cycloplegic conditions, children who had started 0.125% atropine AMI treatment between January and July 2021, and children reporting for their 1-year follow-up visit after treatment completion.

The exclusion criteria excluded patients with anisometropia over 2 diopters under cycloplegic conditions, myopia-related retinal dystrophies, or collagen syndromes and development disorders; patients who used orthokeratology (OK) lenses, other concentrations of atropine, or other methods to treat myopia; and patients with only one eye that met the criteria. This study was conducted in accordance with the Declaration of Helsinki of 1975, as revised in 1983, and was approved by the Korean National Institute for Bioethics Policy (approval number: P01-202206-01-009, date of approval: 6 June 2022). The requirement of written informed consent was waived by the Korean National Institute for Bioethics Policy due to the study’s retrospective nature.

Between January and July 2021, 71 consecutive children with progressive myopia (SE progression rate ≥ 1 D/year under cycloplegic conditions) visited our clinic and were considered eligible for inclusion in this study. Among them, 71 patients (142 eyes) with myopia progression were enrolled in the study and received alternating monocular 0.125% atropine instillation treatment (Myoguard^®^, Light Pharm Tech, Ltd., Seoul, Republic of Korea). In the treatment group, one eye a day is assessed, and the opposite eye is assessed on the following day. In total, 60 patients (120 eyes) who completed one year of alternating monocular instillation (AMI) treatment with a the last follow-up between January and July 2022 and good compliance were allocated to the treatment group. In total, 11 patients (22 eyes) quit treatment within 3 weeks of initiation due to poor compliance with the parents’ interview. These patients showed no rebound effects and were excluded from this study. Between January and July 2021, we reviewed charts of children treated for myopia. A total of 60 patients with the same age, SE, and axial length as the treatment group who did not receive any treatments except for wearing single vision glasses for myopia were allocated to the control group.

A standardized scheduled ophthalmological examination on the both eye was performed at baseline and 1 month, 4 months, and 12 months post-treatment. The best corrected and uncorrected visual acuities were assessed using the 6 m visual acuity chart (Han’s vision chart of a logarithmic chart in Republic of Korea). Cycloplegic refraction was assessed 45 min after the instillation of 1% cyclopentolate eye drops. During follow-up, cycloplegia was already present at examination due to the use of atropine; this was confirmed by the investigators using dynamic retinoscopy and considered a measure of compliance with a Topcon KR-8900 Autorefractor (Topcon, Tokyo, Japan). The pupil size and AL were measured with an optical biometer AL-Scan (NIDEK, Aichi, Japan) approximately 12 h after the instillation of 0.125% atropine during each follow-up visit under the same illumination. Owing to the rapid measurement and good correlation with the Goldman applanation tonometer [25], we used a Topcon CT-80 computerized tonometer (Topcon, Tokyo, Japan) to measure the intraocular pressure (IOP) of the participants. SE was calculated using the standard formula (SE = sphere + 1/2 cylinder). The corneal endothelial cell count was measured using a specular microscope SP-2000P (Topcon, Tokyo, Japan).

The annual myopia progression rate before treatment was calculated by subtracting the sphere at baseline from the sphere estimated one year before treatment for each patient, and the annual myopia progression rate post-treatment was calculated by subtracting the sphere at baseline from the sphere estimated 1-year post-treatment for each patient.

Absolute changes in ocular parameters such as refractive error and the axial length of both groups over the course of 1 year were compared using independent *t*-tests. The SE and AL one year before initiating atropine treatment were compared with those one year after initiating treatment in the treatment group using paired *t*-tests. The incidence of adverse events at baseline and 1 and 12 months after initiating atropine instillation was also evaluated. All statistical analyses were performed using SPSS version 25.0 (IBM Corp., Armonk, NY, USA). *p*-values < 0.05 were considered statistically significant.

### Patient and Public Involvement

Patients or the public were not involved in the design, conduct, reporting, or dissemination plans of our research.

## 3. Results

A total of 120 patients (240 eyes) with myopia progression were enrolled in the study, including 60 patients (120 eyes) who received the alternating monocular 0.125% atropine instillation treatment (Myoguard^®^, Light Pharm Tech, Ltd., Seoul, Republic of Korea) with glasses for one year (treatment group) and 60 patients (120 eyes) of the same age, SE, and axial length as the treatment group, who received no other treatments for myopia except eye glasses and were followed up for one year (control group).

Table 1 presents the characteristics of the two groups at baseline and one year after alternating monocular 0.125% atropine instillation treatment. There were no significant differences in age, SE, and AL between the two groups at baseline (*p* = 0.452; *p* = 0.654; *p* = 0.673, respectively); however, the increases in SE and AL after one year were significantly less in the treatment group than that in the control group (*p*s < 0.001). The annual myopia progression rates before treatment were the same of 37.0% both group; however, the rates observed 12 months later were enhanced in the treatment group (0.36 D/y, 9.1%) compared with the control group (1.02 D/y, 26.4%) (*p*< 0.001) (Table 1).

Moreover, in the treatment group, the change in SE one year before starting treatment was −1.12 ± 0.11 D and was significantly reduced to −0.36 ± 0.46 D after one year of treatment (*p* < 0.001). Similarly, the change in AL at baseline was 0.65 ± 0.18 mm, which reduced to 0.21 ± 0.20 mm after one year of treatment (*p* < 0.001). In contrast, the changes in SE and AL in the control group before one year and after one year of treatment were not different (*p* = 0.786, *p* = 0.651, respectively) (Table 2, Figure 1 and Figure 2). In the treatment group, the differences in SE, sphere, and AL change after one year of treatment were −0.36 ± 0.46 D, −0.30 ± 0.42 D, and 0.21 ± 0.20 mm, respectively. However, the IOP and endothelial cell counts were within normal limits at baseline and after one year of treatment. The pupil size increased by 0.03 ± 0.05 mm and 0.76 ± 0.90 mm under mesopic and photopic conditions, respectively, in the treatment group (Table 3 and Figure 2).

The change in SE one year before starting the treatment was significantly reduced from −1.12 ± 0.87 to −0.36 ± 0.46 D in the treatment group, but the change was from −1.12 ± 0.87 to −1.02 ± 0.57 in the control group after one year of treatment (*p* < 0.001). Similarly, the change in AL at baseline was reduced from 0.65 ± 0.19 to 0.21 ± 0.20 mm in the treatment group and from 0.65 ± 0.19 to 0.51 ± 0.20 in the control group after one year of treatment (*p* < 0.001).

In the treatment group, two patients (3.3%) complained a mild side effect (a tingling sensation) within three months of treatment. After that, no complains of tingling sensation or other side effects, such as photophobia, disturbed near vision, headache, allergic reaction, or systemic flushes, were reported, so compliance was excellent (Table 4).

## 4. Discussion

The AMI method was selected in this study because in two studies of 0.125% atropine binocular instillation, 4.05% and 74% of patients, separately, experienced pupil dilation, photophobia, and disturbed near vision during binocular 0.125% atropine instillation treatment [28,29]. In November 2020, the Korean Ministry of Food and Drug Safety of the Republic of Korea approved the use of 0.125% low-concentration atropine (only one dosage). According to a previous report, the optimal dose for low-concentration atropine eye drops is 0.05% [21]. We hypothesized that the atropine dose could be reduced to a concentration lower than 0.125%, similar to the optimal dose, using the AMI approach.

The change in SE was significantly reduced over the one-year course of treatment. Although age and SE differed across reports, this result may be similar to the one-year SE changes in −0.38 ± 0.35 D/year and −0.47 ± 0.45 D/year with 0.01% and 0.02% atropine eye drops, respectively [26]. With 0.125% atropine eye drops, Lan CS et al. [28] reported a −0.38 ± 0.36 D SE change/year and 0.23 ± 0.19 mm axial length change/year with binocular instillation, which was very similar to that noted in our study (−0.36 ± 0.46 D and 0.21 ± 0.20 mm, respectively) with AMI. Compared with the phase 1 LAMP study [30], the 1-year SE changes of −0.36 ± 0.46 D with AMI of 0.125% atropine in our study were similar to the values of −0.27 ± 0.61 D and −0.46 ± 0.45 D obtained with daily binocular instillation of 0.05% and 0.025% atropine reported.

Additionally, since the medication is instilled in only one eye once a day before going to bed, it may enable better compliance in children compared with protocols where instillation in both eyes is required.

High-concentration atropine eye drops have been associated with several side effects. The most common side effects include photophobia and altered near vision [19,20,21]. However, since the concentration of 0.125% atropine used in this study was also relatively high, the side effects of atropine eye drops were reduced by alternating instillation between eyes. To the best of our knowledge, this is the first instance of its use for pediatric myopia. In this study, only 3.3% complained of a tingling sensation, and this rate is similar to that previously reported for eye irritation [28] and probably due to the weak acidic nature of 0.125% atropine eye solution (Myoguard^®^, Light Pharm Tech, Ltd., Seoul, Republic of Korea).

The pupil size under photopic conditions before atropine instillation was 4.62 ± 0.25 mm in this study; after one year of treatment, the pupil size under photopic conditions increased from 4.62 ± 0.25 mm to 5.50 ± 0.58 mm. There were no complaints of photophobia. Pupil sized of 6.60 ± 1.01 mm for the right eye and 6.64 ± 1.01 for the left eye at one year post-treatment have been reported with binocular instillation of 0.125% atropine [29]. A previous study reported the use of 0.125% atropine eye drops in the myopic eye of anisometropic patients with myopia in one eye and hyperopia in the other eye to reduce the AL difference between eyes, thereby improving the anisometropia [31]. However, studies on the instillation of 0.125% atropine eye drops in both eyes are rare [28,29].

Atropine treatment for myopia progression is used as the standard of care in Taiwan [29] and other countries. Since its approval for use in Korea, it is expected to become the standard of care for Korean pediatric myopia patients.

Although the mechanism through which atropine inhibits myopia progression in children is unclear, it is thought that atropine increases dopamine secretion by acting on the retinal muscarinic receptors [32]. In Northeast Asia, the incidence of myopia in children is very high, and myopia progresses frequently. Since AMI of 0.125% atropine has fewer side effects, it may be more suitable to start atropine eye drop treatment at a SE ≤ −1.0 D or less, which is the initial stage of myopia progression.

There are several limitations to our study. First, our study did not investigate the differences between alternating monocular and binocular eye instillation of 0.125% atropine eye drops. However, it should be noted that AMI of 0.125% atropine eye drops may lead to fewer side effects and better cooperation. Second, the study had a relatively small sample size and a short follow-up period. Long-term effects may become apparent with a longer period of follow-up. Third, this study only included a single race of pediatric patients (Korean). Fourth, the retrospective design of this study is a limitation. Fifth, light exposure and outdoor activity among the participants were not assessed in patients.

In conclusion, alternating 0.125% atropine monocular instillation treatment effectively inhibited myopia progression in children, with minimal side effects. Thus, it can be considered a treatment option for children at risk of developing high myopia.

## Figures and Tables

**Figure 1 jcm-13-05003-f001:**
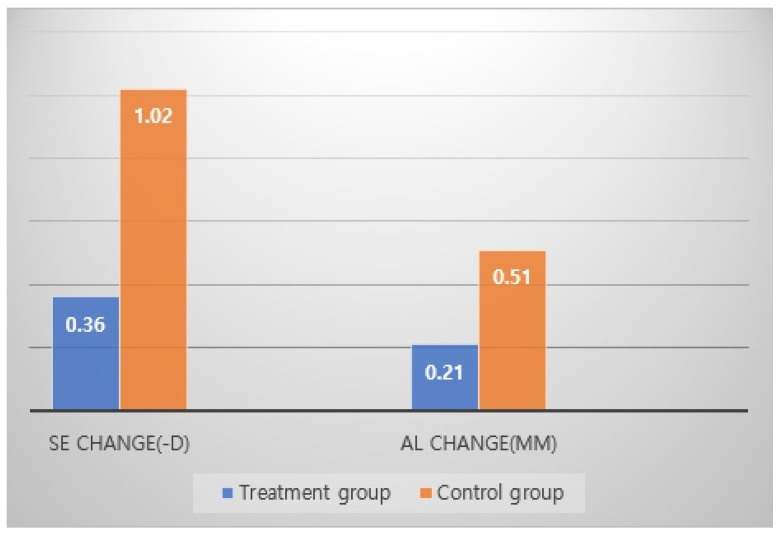
Efficacy of treatment with AMI of 0.125% atropine after one year.

**Figure 2 jcm-13-05003-f002:**
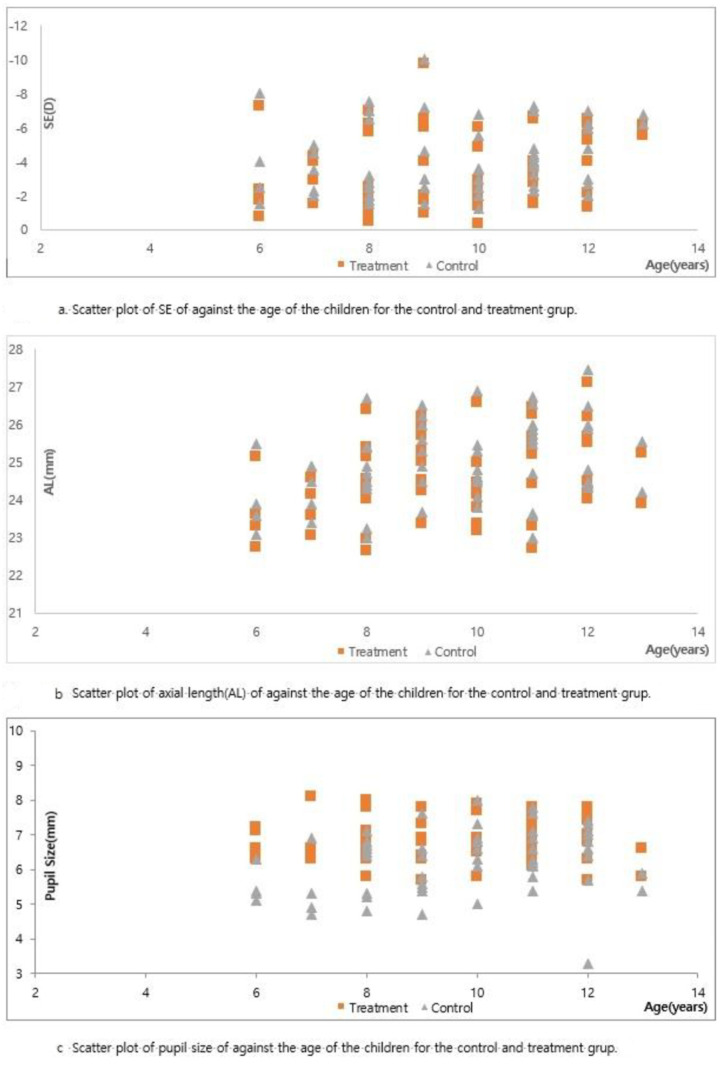
Scatter plot of distributions of SE (**a**), AL (**b**), and pupil size (**c**) in the treatment and control groups.

**Table 1 jcm-13-05003-t001:** Comparison between the treatment (AMI) and the control groups.

	Treatment Group (*n* = 60)	Control Group (*n* = 60)	*p*-Value
Baseline			
Age (years)	9.2 ± 2.0	9.2 ± 1.9	0.452
SE (D)	−3.90 ± 1.56	−3.87 ± 1.55	0.654
AL (mm)	24.68 ± 1.14	24.65 ± 1.20	0.673
12 months later			
SE change/year (D)	0.36 ± 0.46	1.02 ± 0.57	<0.001
AL change/year (mm)	0.21 ± 0.20	0.51 ± 0.20	<0.001
Annual myopia progression rate			
Before treatment	37.0%	37.0%	
12 months later	9.1%	26.4%	<0.001

Abbreviations: AL, axial length; AMI, alternating monocular instillation; D, diopters; SE, spherical equivalent.

**Table 2 jcm-13-05003-t002:** Comparison of spherical equivalent and axial length changes one year before and after treatment.

	One Year before Treatment	One Year after Treatment	*p*-Value
Treatment group			
SE change (D)	−1.12 ± 0.11	−0.36 ± 0.46	<0.001
AL change (mm)	0.65 ± 0.18	0.21 ± 0.20	<0.001
Control group			
SE change (D)	−1.13 ± 0.12	−1.02 ± 0.57	0.786
AL change (mm)	0.65 ± 0.19	0.51 ± 0.20	0.651

Abbreviations: D, diopters; SE, spherical equivalent; AL, axial length.

**Table 3 jcm-13-05003-t003:** Ophthalmological examination findings at baseline and one year after alternating monocular instillation treatment in the treatment group.

		Baseline	1 Year after AMI Treatment	Changes during the Year	*p*-Value
Spherical equivalent (D)		−3.90 ± 1.56	−4.23 ± 1.28	−0.36 ± 0.46	<0.001
Sphere (D)		−3.22 ± 1.49	−3.49 ± 1.21	−0.30 ± 0.42	<0.001
Axial length (mm)		24.68 ± 1.14	24.89 ± 1.47	0.21 ± 0.20	<0.001
IOP (mmHg)		13.90 ± 3.21	13.88 ± 2.25	0.02 ± 0.98	0.567
Pupil size (mm)					
	Mesopic	7.05 ± 0.39	7.08 ± 0.64	0.03 ± 0.05	0.476
	Photopic	4.62 ± 0.25	5.50 ± 0.58	0.76 ± 0.90	0.032
Endothelial cell counts (/mm^2^)		3531 ± 489.7	3446 ± 477.7	−84.6 ± 685.3	0.782

Abbreviations: AMI, alternating monocular instillation; D, diopters; IOP, intraocular pressure.

**Table 4 jcm-13-05003-t004:** Adverse events in the treatment group (AMI).

	Treatment Group (*n*, %)
Tingling sensation	2/60(3.3)
Photophobia	0
Reading problems	0
Systemic (headache, flushes)	0
Infections (blepharitis, conjunctivitis)	0

Abbreviations: AMI, alternating monocular instillation.

## Data Availability

The supporting data for the findings of this study are available on request from the corresponding author, B.M.M. The data are not publicly available due to restrictions as the data may contain information that could compromise the privacy of research participants.

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
