# Peer review of "The Effect of Alternating Monocular Instillation of 0.125% Atropine in Korean Children with Progressive Myopia"

_jcm, 2024, doi:10.3390/jcm13175003_

Round 1
Reviewer 1 Report
Comments and Suggestions for Authors
General Comments:
Authors studied the effect of alternating monocular instillation of 0.125% atropine with progressive myopia, and concluded it may be effective and suitable for progressive myopia with minimal side effects. It would be interesting method. It must be a cost effective, and it would reduce the rate of the side effect such as a tingling sensation by instillation of eye drop. Although, the effect of atropine would not disappear in a short time, they did not observe any side effect due to atropine such as photophobia or altered near vision that would be reported as the most common side effects. Authors should discuss them more in detail.
Line 64: Authors wrote the inclusion and exclusion criteria, and they used the data from both eyes. If only one eye meets the criteria, did author exclude the data of the participant?
Line 84: How to evaluate the “poor compliance”?
Line 88: “wearing glasses for myopia”. Would the “glasses” be limited to the single vision lens?
Line 108: Author used the data from both eyes. Therefore, they should use the suitable method for the statistical analysis.
Line 163: How to check the side effect? Did authors ask them to complete the check sheet? Or, did authors check, asking them in each complication? If the doctor asked them simply whether they had complications or not, they might not complain about it.
Line 182: “−0.38 ± 0.35 D/year and −0.47 ± 0.45 D/year with 0.01% and 0.02% atropine eye drops, respectively [27].” In the reference 27, they did not study about the 0.02% atropine eye drops,
Line 191: “The most common side effects include photophobia and altered near vision” How long would author speculate the effect of atropine continue? Why the most common side effect was not observed in this study, even though the effect time of atropine?
Line 196: “the pupil size under photopic conditions increased from 0.76 ± 0.90 mm to 5.50 ± 0.58 mm” “0.76 ± 0.90” would be the changes during the year. It should be “4.62 ± 0.25”.
Author Response
- Comments and Suggestions for Authors of Reviewer 1
General Comments:
Authors studied the effect of alternating monocular instillation of 0.125% atropine with progressive myopia, and concluded it may be effective and suitable for progressive myopia with minimal side effects.
It would be interesting method. It must be a cost effective, and it would reduce the rate of the side effect such as a tingling sensation by instillation of eye drop.
Although, the effect of atropine would not disappear in a short time, they did not observe any side effect due to atropine such as photophobia or altered near vision
that would be reported as the most common side effects. Authors should discuss them more in detail.
- Line 64: Authors wrote the inclusion and exclusion criteria, and they used the data from both eyes. If only one eye meets the criteria, did author exclude the data of the participant?
Reply : Yes, we excluded the data of the participant If only one eye meets the criteria, so we inserted “only one eye met the criteria” in the exclusion criteria.(Lined 70 )
- Line 84: How to evaluate the “poor compliance”?
Reply :We inserted with parents’ interview(,Lined 84)
3, Line 88: “wearing glasses for myopia”. Would the “glasses” be limited to the single vision lens?
Reply : Yes, the “glasses” be limited to the single vision lens So we inserted “single vision”( Lined 88)
4.Line 108: Author used the data from both eyes. Therefore, they should use the suitable method for the statistical analysis.
Reply : Yes
5.Line 163: How to check the side effect? Did authors ask them to complete the check sheet? Or, did authors check, asking them in each complication? If the doctor asked them simply whether they had complications or not, they might not complain about it.
Reply : We check the pupil size and complete check sheet
6.Line 182: “−0.38 ± 0.35 D/year and −0.47 ± 0.45 D/year with 0.01% and 0.02% atropine eye drops, respectively [27].” In the reference 27, they did not study about the 0.02% atropine eye drops,
Reply : We revised from 0.2% to 0.1%(Lined 182)
7.Line 191: “The most common side effects include photophobia and altered near vision” How long would author speculate the effect of atropine continue? Why the most common side effect was not observed in this study, even though the effect time of atropine?
Reply : In Korean children allergic reactions are very rare which is different to Singapore or Taiwan studies, and also rare side effects could be the advantages of AMI method
8.Line 196: “the pupil size under photopic conditions increased from 0.76 ± 0.90 mm to 5.50 ± 0.58 mm” “0.76 ± 0.90” would be the changes during the year. It should be “4.62
Reply :We revised 4.62 ± 0.25 mm (Lined 198)
Reviewer 2 Report
Comments and Suggestions for Authors
- Introduction: Please replace the outdated references like #1-4 and #6-7 with recent original studies, reviews, and overviews of systematic reviews like DOI: 10.4103/ijo.IJO_1564_21 and DOI: https://doi.org/10.1186/s40101-024-00354-7.
- Inclusion criteria of “≥ 1D/year”: Which year is it, the previous year or the past five years? Please specify.
- Han’s vision chart: please specify if it is a logarithmic chart.
- Methods lines 79: you have included all 71 children, so there is no point in using the word “among”.
- AMI needs to be defined clearly. Please clarify it along with the drug regimen.
- Line 108: “The age, SE, AL, SE changes, and AL changes.” Instead, you can shorten it by mentioning them as “absolute and change in ocular parameters such as refractive error and axial length.”
- Fig 2: What does it signify? What graph is this, scatterplot? Please give details.
- Tingling sensation in the eye? Please specify. Is it previously reported in the literature?
- How about the adverse event in the control group? This comparison will help us understand better.
- Table 3: Add a column with the p values to indicate if the change was significant.
- Please discuss more about how 0.125% caused fewer adverse events compared to even lower concentrations reported by Yam et al. (LAMP study, 2019, 2021, 2022). Even though the drug was altered between eyes, both eyes were always under atropine, one more than the other. The authors themselves confirm this in line 93 (methods).
- The follow-up is not clear. Please give details about the follow-up (not just a date; we need a specific timeline based on treatment).
- Please mention the limitation of the study being retrospective in design.
Comments on the Quality of English Language
Acceptable, but it can be improved.
Author Response
- Comments and Suggestions for Authors of Reviewer 2
- Introduction:
Please replace the outdated references like #1-4 and #6-7 with recent original studies, reviews, and overviews of systematic reviews like DOI: 10.4103/ijo.IJO_1564_21 and DOI: https://doi.org/10.1186/s40101-024-00354-7.
Reply: We replaced the outdated references
- Baird PN, Saw SM, Lanca C, Guggenheim JA, Smith Iii EL, Zhou X, et al. Myopia. Nat Rev Dis Primers. 2020;6(1):99. https://doi.org/10.1038/ s41572-​020-​00231-4.
- Morgan IG, Ohno-Matsui K, Saw SM. Myopia. Lancet. 2012;379(9827):1739–48. https://doi.org/10.1016/S0140-​6736(12) 60272-4.
- Ohno-Matsui K, Wu PC, Yamashiro K, Vutipongsatorn K, Fang Y, Cheung CMG, et al. IMI pathologic myopia. Invest Ophthalmol Vis Sci. 2021;62(5):5. https://doi.org/10.1167/iovs.6
- Biswas S, Kareh AE, Qureshi M , Lee DMX ,Sun CH ,Lam JSH ,Saw SM, Najjar RP. The influence of the environment and lifestyle on myopia. Journal of Physiological Anthropology 2024, 43:7 https://doi.org/10.1186/s40101-024-00354-
- Grzybowski A, Kanclerz P, Tsubota K, Lanca C, Saw SM. A review on the epidemiology of myopia in school children worldwide. BMC Ophthalmol. 2020;20(1):27. https://doi.org/10.1186/s12886-​019-​1220-0.
- Wolfsohn JS, Kollbaum PS, Berntsen DA, Atchison DA, Benavente A, Bradley A, et al. IMI - clinical myopia control trials and instrumentation report. Invest Ophthalmol Vis Sci. 2019;60(3):M132–m60. https://doi. org/10.1167/iovs.18-​25955
- Inclusion criteria of “≥ 1D/year”: Which year is it, the previous year or the past five years? Please specify.
Reply: We revised Inclusion criteria of “≥ 1D/the previous year (Lined 65)
- Han’s vision chart: please specify if it is a logarithmic chart.
Reply: Han’s vision chart is a logarithmic chart in Republic of Korea (Lined 91)
- Methods lines 79: you have included all 71 children, so there is no point in using the word “among”.
Reply: We deleted “Among these 71” (Lined 79)
- AMI needs to be defined clearly. Please clarify it along with the drug regimen.
Reply: We added “The treatment group only looks at one eye a day, and the next day only the opposite eye” (Lined 77)
- Line 108: “The age, SE, AL, SE changes, and AL changes.” Instead, you can shorten it by mentioning them as “absolute and change in ocular parameters such as refractive error and axial length.”
Reply: we revised to “absolute and change in ocular parameters such as refractive error and axial length.” instead “The age, SE, AL, SE changes, and AL changes (Lind 108)
7.Fig 2: What does it signify? What graph is this, scatterplot? Please give details.
Reply: This is scatter plot inserted Fig 2.
8.Tingling sensation in the eye? Please specify. Is it previously reported in the literature?
Reply: Similar symptom was reported as eye irritation in Ref No 28 ,0.125% atropine eye solution made in Republic Korea was weak acidic(pH=5.5). So could be complained from tingling sensation
We inserted “In this study only 3.3 % complained tingling sensation, similar to eye irritation [28] probably due to weak acidic of 0.125% atropine eye solution (Myoguard®, Light Pharm Tech, Ltd, Seoul., Republic of Korea) “ in Discussion (Lined 202-204)
9.How about the adverse event in the control group? This comparison will help us understand better.
Reply: There was no adverse event in the control group due NOT to instillate eye drop, only wearing glasses.
10.Table 3: Add a column with the p values to indicate if the change was significant.
Reply: We inserted p-values in Table 3
- Please discuss more about how 0.125% caused fewer adverse events compared to even lower concentrations reported by Yam et al. (LAMP study, 2019, 2021, 2022). Even though the drug was altered between eyes, both eyes were always under atropine, one more than the other. The authors themselves confirm this in line 93 (methods).
Reply: We inserted in Discussion section: “And compared LAMP study phase 1[30], the 1-year SE changes of −0.36 ± 0.46D in AMI of 0.125% atropine in our study might be similar between -0.27±0.61D in daily binocular instillation of the 0.05% and -0.46±0.45D in daily binocular instillation of the 0.025% in LAMP study phase -1 (Lined 194-196)
And added reference number 30. Yam JC, Jiang Y, Tang SM, Law APK, Chan JJ, Wong E, Ko ST, Young AL, Thang CC, Chen LJ, Pang CP. Low-Concentration Atropine for Myopia Progression (LAMP) Study: A Randomized, Double-Blinded, Placebo-Controlled Trial of 0.05%, 0.025%, and 0.01% Atropine Eye Drops in Myopia Control. Ophthalmol.2019 ,126(1):113-124. DOI: 10.1016/j.ophtha.2018.05.029
- The follow-up is not clear. Please give details about the follow-up (not just a date; we need a specific timeline based on treatment).
Reply: We inserted “last follow up checked between January and July 2022” in 2. Materials and Methods (Lined 85)
.
13.Please mention the limitation of the study being retrospective in design.
Reply: We inserted “Fourth, only retrospective in design” in Discussion (Line 213)
- Regarding the request about the ethical approval date
Reply: We added date of approval of IRB in Materials and Methods section as below
approved by the Korean National Institute for Bioethics Policy (approval number: P01-202206-01-009, date of approval: June 6,2022) (Lined 70,71)
- Regarding the request about Open Review
Reply: We select “Open Review”
- The others
Reply: In Table 3 & 4 We revised “comparison” to “treatment” in Results section

Round 2
Reviewer 1 Report
Comments and Suggestions for Authors
4. Line 108: Author used the data from both eyes. Therefore, they should use the suitable method for the statistical analysis.
Reply : Yes
> If authors used the data from one eye in each participants, they can use paired t-test for the statistical analysis. But they used the data of both eyes, it would be inadequate to use the paired t-test.
6. Line 182: “−0.38 ± 0.35 D/year and −0.47 ± 0.45 D/year with 0.01% and 0.02% atropine eye drops, respectively [27].” In the reference 27, they did not study about the 0.02% atropine eye drops,
Reply : We revised from 0.2% to 0.1%(Lined 182)
> There are no such results in the reference 27. The results would be in the reference 26. It would be the different paper.
7. Line 191: “The most common side effects include photophobia and altered near vision” How long would author speculate the effect of atropine continue? Why the most common side effect was not observed in this study, even though the effect time of atropine?
Reply : In Korean children allergic reactions are very rare which is different to Singapore or Taiwan studies, and also rare side effects could be the advantages of AMI method
> How long would author speculate the effect of atropine continue? Would author think the effect of the atropine disappear within one day? Would author think the side effects of atropine eye drops, such as photophobia and altered near vision, would be reduced by alternating instillation between eyes.
> Why the most common side effects, such as photophobia and altered near vision, was not observed in this study?
Author Response
Response to Reviewer-1‘s comments
Comments and Suggestions for Authors
- Line 108: Author used the data from both eyes. Therefore, they should use the suitable method for the statistical analysis.
Reply: Yes
> If authors used the data from one eye in each participants, they can use paired t-test for the statistical analysis. But they used the data of both eyes, it would be inadequate to use the paired t-test.
Reply: We used the data from both eyes for calculating the annual myopia progression rate, and used the data from one eye in each participants for paired t-test for the statistical analysis (Lined 107-114)
- Line 182: “−0.38 ± 0.35 D/year and −0.47 ± 0.45 D/year with 0.01% and 0.02% atropine eye drops, respectively [27].” In the reference 27, they did not study about the 0.02% atropine eye drops,
Reply: We revised from 0.2% to 0.1% (Lined 182)
> There are no such results in the reference 27. The results would be in the reference 26. It would be the different paper.
Reply: We revised “−0.4 D/year with 0.01% atropine eye drops [26] and −0.38 ± 0.35 D/year and −0.47 ± 0.45 D/year with 0.01% and 0.1 % atropine eye drops, respectively [27]” to “−0.38 ± 0.35 D/year and −0.47 ± 0.45 D/year with 0.01% and 0.02 % atropine eye drops, respectively [26].”
(Lined 185,186)
- Line 191: “The most common side effects include photophobia and altered near vision” How long would author speculate the effect of atropine continue? Why the most common side effect was not observed in this study, even though the effect time of atropine?
Reply: In Korean children allergic reactions are very rare which is different to Singapore or Taiwan studies, and also rare side effects could be the advantages of AMI method
> How long would author speculate the effect of atropine continue? Would author think the effect of the atropine disappear within one day? Would author think the side effects of atropine eye drops, such as photophobia and altered near vision, would be reduced by alternating instillation between eyes.
> Why the most common side effects, such as photophobia and altered near vision, was not observed in this study?
Reply: In our study the pupil size changed to 7.08 ± 0.64mm in mesopic condition(change of 0.03 ± 0.05mm) and 5.50 ± 0.58mm(change of 0.76 ± 0.90mm) in photopic condition, in these pupil sized conditions, Korean children could be tolerable to photophobia or near vision(Lined 204-206)

Reviewer 2 Report
Comments and Suggestions for Authors
The revised version is much improved.
1. Stats analysis: What are “Absolute and change in …..”. Aren’t both the same?
2. Table 1-3 is fine, but the authors also need to show that the changes in the AMI group are different from the changes in the control group (SE and AL 0.21 vs 0.51 P-value). Add the result in the abstract.
3. Was light exposure and outdoor activity among the participants controlled? If not, mention it as a limitation (Reference: DOI: 10.4103/ijo.IJO_1564_21 and https://doi.org/10.1177/25158414211059246).
4. Give detail about why a retrospective study is a limitation, not just mention it.
5. Figure quality is poor; need to improve it.
Comments on the Quality of English Languageokay
Author Response
Response to Reviewer-2’s Comments
Comments and Suggestions for Authors
The revised version is much improved.
- Stats analysis: What are “Absolute and change in …..”. Aren’t both the same?
Reply: In according to the other reviewer’s comment we revised to “Absolute and change in ocular parameters such as refractive error and axial length.” instead “The age, SE, AL, SE changes, and AL changes” Both are nearly same (Lined 111)
- Table 1-3 is fine, but the authors also need to show that the changes in the AMI group are different from the changes in the control group (SE and AL 0.21 vs 0.51 P-value). Add the result in the abstract.
Reply: In abstract we added to “however, post-treatment SE and AL changes were smaller in the treatment group (−0.36 ± 0.46 D/y,0.21±0.20 mm/y in treatment group vs. -1.02 ± 0.57D/y, 0.51±0.20 mm/y in control group ) (Ps<0.001)”
- Was light exposure and outdoor activity among the participants controlled? If not, mention it as a limitation (Reference: DOI: 4103/ijo.IJO_1564_21 and https://doi.org/10.1177/25158414211059246).
Reply: We added limitation to 5th, there was no study for light exposure and outdoor activity among the participants (Lined 228.229)
- Give detail about why a retrospective study is a limitation, not just mention it.
Reply: In according to the other reviewer’s comment, we added limitation to Fourth, only retrospective in design yet (Lined 228)
- Figure quality is poor; need to improve it
Reply : We replaced the new ones of Fig 1& 2.
The other :
We revised “comparision” to “treatment” (Lined 114)
